# E3 Ubiquitin Ligase ASB17 Promotes Apoptosis by Ubiquitylating and Degrading BCLW and MCL1

**DOI:** 10.3390/biology10030234

**Published:** 2021-03-18

**Authors:** Ge Yang, Pin Wan, Qi Xiang, Shanyu Huang, Siyu Huang, Jun Wang, Kailang Wu, Jianguo Wu

**Affiliations:** 1State Key Laboratory of Virology, College of Life Sciences, Wuhan University, Wuhan 430072, China; 2016202040041@whu.edu.cn (G.Y.); 2017202040043@whu.edu.cn (Q.X.); 2018202040014@whu.edu.cn (S.H.); 2015202040054@whu.edu.cn (S.H.); 2Guangdong Provincial Key Laboratory of Virology, Institute of Medical Microbiology, Jinan University, Guangzhou 510632, China; 2019268wp@stu.jnu.edu.cn; 3Affiliated Shunde Hospital of Jinan University, Foshan 528305, China; Yxh19830924@163.com; 4Foshan Institute of Medical Microbiology, Foshan 528315, China

**Keywords:** ankyrin repeat and SOCS box-containing 17, ASB17, apoptosis, etoposide, B-cell leukemia/lymphoma w, BCLW, caspases, myeloid cell leukemia-1, MCL1

## Abstract

**Simple Summary:**

B-cell lymphoma-2 family proteins have been widely accepted as the critical regulators in cell apoptosis, often found to be abnormally expressed in many cancers. Among them, B-cell leukemia/lymphoma w and myeloid cell leukemia-1 are two pro-survival proteins. Here, we reported that the ankyrin repeat and SOCS box protein 17 can degrade the two proteins in a ubiquitylation -dependent way. Furthermore, we generated the first ASB17 knockout C57BL/6J mice line. The results revealed that ASB17 deficiency inhibited apoptosis but did not affect testes development. Moreover, the ASB17-deficient mice were more resistant to the stimuli of etoposide, Altogether, these findings indicate that ASB17 is a novel positive mediator of cell apoptosis.

**Abstract:**

Apoptosis is a very important process of cell death controlled by multiple genes during which cells undergo certain events before dying. Apoptosis helps to clean the unnecessary cells and has critical physiological significance. Altered apoptosis results in a disorder of cell death and is associated with many diseases such as neurodegenerative diseases and cancers. Here, we reported that the ankyrin repeat and SOCS box protein 17 (ASB17) was mainly expressed in the testis and promoted apoptosis both in vivo and in vitro. Analyzing ASB17-deficient mice generated by using the CRISPR/Cas9 system, we demonstrated that ASB17 deficiency resulted in the reduction of apoptosis in spermatogenic cells, but it did not affect the development of spermatozoa or normal fertility. Next, in an in vivo model, ASB17 deficiency prevented the apoptosis of spermatogonia induced by etoposide in male mice. We noted that ASB17 promoted apoptosis in a caspase-dependent manner in vitro. Moreover, ASB17 interacted with the members of the BCL2 family, including BCL2, BCLX, BCLW, and MCL1. Interestingly, ASB17 specifically degraded the two anti-apoptotic factors, BCLW and MCL1, in a ubiquitylation-dependent fashion. Collectively, our findings suggested that ASB17 acted as a distinct positive regulator of cell apoptosis.

## 1. Introduction

Apoptosis is a complicated process during which the regulated destruction of a cell happens, and the activity of many genes influences this process [1]. The typical morphological characteristics of apoptosis include cell shrinkage, nuclear fragmentation, chromatin condensation, and membrane blistering [2,3,4]. There are intrinsic and extrinsic pathways of apoptosis [5]. The BCL2 family proteins comprise the network that regulates the mitochondrial or intrinsic apoptotic response and makes it possible to react quickly to specific apoptotic stimuli [6]. The extrinsic apoptotic pathway requires death factors and receptors, including Fas-induced apoptosis and TNF-induced apoptosis [7]. Spermatogenesis is a complicated process in which germline stem cells undergo mitosis, meiosis, and spermiogenesis phases [8]. Apoptosis eliminates excess spermatogonia generated during mitosis to maintain the homeostasis of spermatogenesis [9,10,11,12]. Furthermore, germ cells during meiosis are highly sensitive to DNA damage, heat shock, ionizing radiation, growth factor deprivation, and chemotherapeutic agents [13]. Etoposide, which is widely used against testicular cancer, can induce male germ cell apoptosis [14].

ASB17 is a member of the Ankyrin repeat and SOCS box-containing protein (ASB) family belonging to a family of E3 ubiquitin ligases [15]. It contains two ankyrin repeats and one SOCS box [16,17]. The whole ASB family contains 18 members from ASB1 to ASB18 [18]. Several ASB proteins have been identified to be the component of E3 ubiquitin ligases. ASB2 was found to interact with Cullin5 and Rbx2 to form E3 ubiquitin ligase complexes [18]. ASB7 degrades DDA3 to regulate spindle dynamics and genome integrity [19]. ASB9 targets ubiquitous mitochondrial creatine kinase (uMtCK) and regulates mitochondrial function [20]. ASB11 degrades BIK to influence cell life/death [21]. However, the biological functions of ASB17 in the apoptosis signaling pathway remain unknown.

In this study, we described a distinct mechanism by which ASB17 promoted cell apoptosis. By generating and analyzing ASB17-deficient mice, we determined the biological function of ASB17 in the regulation of the apoptosis signaling pathway. Strikingly, ASB17 deficiency resulted in the reduction of apoptotic cell apoptosis in spermatogenic cells, but it did not affect the development of spermatozoa. In an in vivo model, ASB17 deficiency prevented the apoptosis of spermatogonia induced by etoposide in male mice. Furthermore, ASB17 overexpression led to the promotion of apoptosis in cultured cells. ASB17 could bind to BCLW and MCL1 and promote the ubiquitylation and degradation of BCLW and MCL1 in cultured cells. Therefore, ASB17 positively regulated cell apoptosis by promoting the ubiquitylation and degradation of BCLW and MCL1.

## 2. Materials and Methods

### 2.1. Animal Studies

C57BL/6J ASB17^+/−^ mice were generated by Beijing Biocytogen (Beijing, China) using the CRISPR/Cas9 technique. All animal experiments were undertaken in accordance with the National Institutes of Health Guide for the Care and Use of Laboratory Animals, with the approval of the Wuhan University Animal Care and Use Committee guidelines. Two pairs of primers were used to examine the genotypes of mice: WT-F: 5′-GGCTGCATCACTCTGCCGCT-3′; WT-R: 5′-TCCGAAGGCAACAGAAGGGGAA-3′; Mut-F: 5′-AGGAAGCCCCAGGTCTTCATCCT-3′.

### 2.2. Intratesticular Injections

Male mice of 6 weeks were anesthetized with avertin. The testes were exteriorized through a low midline abdominal incision. Ten microliters of a solution containing 80 mM etoposide in DMSO were injected into the testes via a 30G needle. Following injection, the testes were returned to the peritoneal cavity, and the incisions were sutured. The mice were recovered and raised for 24 h. DMSO was injected into the testes as a control.

### 2.3. Cell Lines and Cultures

HEK293T cells, Hela cells, spermatogonium GC1-spg cells, spermatocyte GC2-spd cells, Leydig (TM3) cells, and Sertoli (TM4) cells were purchased from American Type Culture Collection (ATCC) (Manassas, VA, USA). HEK293T and Hela cells were cultured in DMEM purchased from Gibco (Grand Island, NY, USA) supplemented with 10% fetal bovine serum (FBS), 100 U/mL of penicillin, and 100 μg/mL of streptomycin sulfate. Spermatogonium GC1-spg, spermatocyte GC2-spd, Leydig (TM3), and Sertoli (TM4) cells were cultured in DMEM/F12 (GIBCO) supplemented with 10% FBS, 100 U/mL of penicillin, and 100 μg/mL of streptomycin sulfate.

### 2.4. Reagents

Staurosporine (STS) (Cat. Number T6680) and Etoposide (ETO) (Cat. Number T0132) were purchased from Target Mol (Shanghai, China). Puromycin (Cat. Number S7417), CHX (Cat. Number S7418), and MG132 (Cat. Number S2619) were purchased from Selleck (Selleckchem, Houston, TX, USA). Polybrene (Cat. Number TR-1003-G) was purchased from Sigma-Aldrich (St. Louis, MO, USA). EDTA-free Protease Inhibitor Cocktail Tablets provided in EASYpacks were purchased from Roche (Indianapolis, IN, USA). Trizol reagent and Lipofectamine 2000 transfection reagent (Cat. Number 11668019) were purchased from Invitrogen (Carlsbad, CA, USA).

### 2.5. Antibodies

Anti-PARP1 (46D11) (Cat. Number 9532), anti-Caspase9 (Cat. Number 9504), and anti-Caspase3 (Cat. Number 9662) were purchased from Cell Signaling Technology (Beverly, MA, USA). Anti-Flag (Cat. Number F3165), and anti-HA (Cat. Number H6908) antibodies were purchased from Sigma-Aldrich. Anti-GAPDH (Cat. Number 60004-1-lg) was purchased from Proteintech (Wuhan, Hubei, China). Anti-Mouse IgG Dylight 649 (Cat. Number A23610) and anti-Rabbit IgG FITC (Cat. Number A22120) were purchased from Abbkine (Wuhan, Hubei, China).

### 2.6. RNA Extraction and Quantitative PCR

Total RNA was isolated with Trizol and reversed transcription using the M-MLV Reverse Transcriptase (Promega) to cDNA for quantitative RT-PCR. The Q-PCR primers were listed as below: hASB17-F: 5′-CTGGGTTTTTGCCAGAAAAGGT-3′; hASB17-R: 5′-TGCCACTTAATGGGCTTGGA-3′; hBCLW-F: 5′-CACCCAGGTCTCCGATGAAC-3′; hBCLW-R: 5′-TTGTTGACACTCTCAGCACAC-3′; hMCL1-F: 5′-AAGAGGCTGGGATGGGTTTGTG-3′; hMCL1-R: 5′-TTGGTGGTGGTGGTGGTTGG-3′; hGAPDH-F: 5′-AAGGCTGTGGGCAAGG-3′; hGAPDH-R: 5′-TGGAGGAGTGGGTGTCG-3′; mASB17-F: 5′-GGGTTTTCGCCAGGAAAGGT-3′; mASB17-R: 5′-CTGTCTTGTCTGAGCCACGTA-3′; mGAPDH-F: 5′-TTCACCACCATGGAGAAGGC-3′; mGAPDH-R: 5′-GGCATCGACTGTGGTCATGA-3′.

### 2.7. Plasmids and Constructions

ASB17, ASB17 (ΔSOCS), ATG7, BCLX, BCLW, MCL1, and BCL2 were cloned into pcDNA3.1(+)-3 × Flag vector. BCL2, ASB17, and ASB17 (ΔSOCS) were cloned into the Myc-His (-) vector. ASB17, BCLW, and MCL1 were cloned into the pCAGGS-HA vector. HA-Ub plasmid was kindly provided by Dr. Bo Zhong of Wuhan University, China. All constructs were confirmed by DNA sequencing.

### 2.8. Western Blot Analysis

Cells were lysed in lysis buffer (50 mM Tris-HCl, pH 7.4, 150 mM NaCl, 1% Triton X-100, 5 mM EDTA, and 10% glycerol) plus protease inhibitors, and the lysates were then subjected to 8% or 10% SDS-PAGE and Western blotting.

### 2.9. Coimmunoprecipitation

Cells were washed with cold PBS and lysed in 1 mL of NP-40 lysis buffer (50 mM Tris-HCl, pH 7.4, 150 mM NaCl, 1% NP-40, 1 mM EDTA, and 5% glycerol) plus protease inhibitors. For immunoprecipitation, lysates were incubated with indicated antibodies at 4℃ overnight, and then with protein A/G-agarose (GE Life Sciences; Piscataway, NJ, USA) for 2 h. After the beads were washed 4–7 times with 1 mL of washing buffer (50 mM Tris-HCl, pH 7.4, 300 mM NaCl, 1% NP-40, 1 mM EDTA, and 5% glycerol), and the immunoprecipitated products were absolved with 50 μL of 2 × SDS loading buffer.

### 2.10. Lentiviral Package and Infection

ASB17 was cloned into pLenti-CMV vector, which was derived from the pLenti-CMV-EGFP vector bought from Addgene to generate pLenti-ASB17. Together with pMD2.G (an envelope plasmid) and psPAX2 (a packaging plasmid), pLenti-ASB17 was transfected into HEK293T cells. After 36 h and 48 h post-transfection, the cell supernatants were collected and then passed through a 0.45 μm filter to remove the cells. Hela cells were infected with harvested supernatants plus 8 μg/mL of Polybrene for 24 h. Hela stably expressing ASB17 cells were identified by Q-PCR.

### 2.11. Immunofluorescence Microscopy

The total cells were washed with PBS and fixed with 4% paraformaldehyde for 15 min. And then they were permeabilized with PBS containing 0.2% Triton X-100 for 5 min and blocked with PBS containing 5% BSA for 1 h at 24 ℃. After being incubated with the indicated antibody at 4 ℃ for 12–16 h, the cells were washed with PBS and then incubated with anti-Mouse IgG Dylight 649 and anti-Rabbit IgG FITC at room temperature for 45–60 min. Cells were incubated with DAPI for 5 min at 37 ℃ after being washed with PBS. Eventually, the cells were analyzed using FluoView FV 1000 (Olympus, Tokyo, Japan).

### 2.12. In Vivo Ubiquitylation Assay

The harvested cells were lysed in 100 μL of SDS lysis buffer (50 mM Tris-HCl, pH 7.4, 150 mM NaCl, 1% NP-40, 1 mM EDTA, 5% glycerol, and 1% SDS). Lysates were diluted with 900 μL of dilution buffer (50 mM Tris-HCl, pH 7.4, 150 mM NaCl, 1% NP-40, 1 mM EDTA, and 5% glycerol) plus protease inhibitors after being heated at 95 ℃ for 5 min. Then, 100 μL of lysates were subjected for immunoblot analysis, and 900 μL of lysates was immunoprecipitated with indicated antibodies. The following experimental procedures were the same with coimmunoprecipitation assays.

### 2.13. TUNEL Assay

Testes and epididymis were isolated from male mice and fixed in Bouin’s solution or 4% paraformaldehyde at 4 °C. TUNEL assay was done by using the In Situ Cell Death Detection Kit, POD (No. 11684817910; Roche Applied Science, Indianapolis, IN, USA), in accordance with the manufacturer’s instructions.

### 2.14. Fertility Evaluation

Male ASB17^+/+^ or ASB17^−/−^ mice were mated to wild-type females at a male/female ratio of 1/2. The litter sizes were analyzed as a measure of fertility.

### 2.15. Statistics

All experiments were independently carried out at least three times. The unpaired two-tailed *t*-tests using GraphPad Prism 5 (San Diego, CA, USA) were used to perform Statistical analysis. Difference was considered statistically significant when *p* < 0.05 (*), *p* < 0.01 (**), ***, *p* < 0.01, *p* < 0.0001 (****).

## 3. Results

### 3.1. ASB17 Is Mainly Expressed in the Testis and Epididymis and Slightly Elevated in Spleen and Lung

To investigate the function of the ASB17 gene *in vivo*, we generated ASB17^+/−^ mice by using the CRISPR-Cas 9 technique to delete exon 1 of the whole genome of the ASB17 gene (Figure 1A). After the breeding of ASB17^+/−^ mice, the mutant, heterozygous, and wild-type (WT) mice were bred. Their offspring were then identified by PCR analysis using DNA isolated from tail snips (Figure 1B). The expression of ASB17 mRNA in the testis from WT and ASB17^−/−^ mice was checked by quantitative PCR (Q-PCR) (Figure 1C). RNA isolated from the heart, liver, spleen, lung, kidney, large intestine, small intestine, testis, and epididymis was used for Q-PCR analysis. The results told us ASB17 was mainly expressed in the testis and epididymis and slightly expressed in the spleen and lung (Figure 1D).

Next, we examined the expression pattern of ASB17 in mice testis. The Q-PCR analysis revealed that ASB17 was expressed from three weeks onwards and reached the top level in around four weeks (Figure 1E). We also checked the ASB17 mRNA expression in four mice testis-derived cell lines, i.e., spermatogonium GC1-spg, spermatocyte GC2-spd, Leydig (TM3), and Sertoli (TM4) cells. The results indicated that ASB17 was barely expressed (Figure 1F). HepG2 cells, Huh7 cells, RD cells, U251 cells, A549 cells, Hela cells, THP1 cells, 293T cells, and Hela cells infected by lentivirus-carrying ASB17 vectors were used to examine the mRNA level of ASB17. The results indicated that ASB17 was barely expressed in these cells (Figure 1G). Furthermore, the expression status of ASB17 was determined in different human testicular cells using the Human Atlas Browser https://humantestisatlas.shinyapps.io/humantestisatlas1/ (accessed on 10 December 2020) [22]. It was concluded that the highest level of ASB17 was expressed in the round and elongated spermatids and differentiating germ cells. Taken together, these data suggested that ASB17^−/−^ mice were successfully generated, and ASB17 expression was in a tissue-specific and temporospatial pattern. The high expression pattern of ASB17 in the testis suggested that ASB17 might play roles in testis development and spermatogenesis.

### 3.2. ASB17 Deficiency in the Testes Attenuated Apoptosis with No Effect on Testes Development

To determine the effect of ASB17 on testis development, the testes from mice of different ages were harvested. The testes morphology in six-week ASB17^−/−^ male mice showed no significant difference, compared to those from wild-type mice (Figure 2A). Testes isolated from male ASB17^−/−^ mice weighed almost as much as those from the age-matched WT mice (Figure 2B). To further understand the function of ASB17 in the testis, terminal deoxynucleotidyl transferase dUTP nick-end labeling (TUNEL) was used. Surprisingly, testes isolated from three-week and six-week ASB17^−/−^ male mice had significantly fewer apoptotic cells than those from the control group, whereas the apoptosis in the testes did not change much between two-week KO mice and WT mice (Figure 2C,E,G). Further analyses showed that apoptosis in three-week and six-week ASB17^−/−^ testes decreased by two-fold (*p* = 0.0044) and eight-fold (*p* = 0.0001), respectively (Figure 2F,H), whereas no significant change can be found between two-week KO and WT mice (Figure 2D). Furthermore, ASB17 deficiency was associated with decreased cleaved-Caspase3 protein expression in the testes from six-week mice (Figure 2I). The role of ASB17 in regulating the apoptosis of testis was strongly related to the expression pattern of ASB17 since it was expressed from three weeks and reached the top level by four weeks. Thus, these results indicated that ASB17 promoted apoptosis in the testis when it was expressed from three weeks of age, and the deletion of ASB17 did not affect testes development.

### 3.3. ASB17^−/−^ Mice Display Normal Spermatogenesis

To find out if ASB17 deficiency affected spermatogenesis, hematoxylin and eosin (H&E) staining and periodic acid–Schiff (PAS) staining were used to examine the histology of testes from two, three, and six weeks. Histology analysis revealed that ASB17^−/−^ testes had no obvious structural change and contained a complete lineage of germ cells compared to WT testes (Figure 3A–F). Furthermore, acrosome morphogenesis did not change in ASB17^−/−^ testes, indicating normal spermiogenesis (Figure 3G,H). Almost the same amounts of spermatid cells could be found in the epididymis cauda of six-week WT and ASB17^−/−^ male mice (Figure 3G,H). Next, the fertility of adult ASB17^−/−^ mice was checked. Unsurprisingly, ASB17^−/−^ male mice were fertile, and the size of the litter showed no significant change with the litter farther from the WT group (Figure 3I). In a word, although ASB17 was highly expressed in the testis, it did not seem to have a role in spermatogenesis under normal conditions.

### 3.4. ASB17 Deficiency Prevents the Apoptosis of Spermatogonia Induced by Etoposide in Male Mice

A previous study showed that the intratesticular injection of etoposide promoted germ cell apoptosis [13]. Etoposide dissolved in DMSO was intratesticular injected into both WT and ASB17^−/−^ male mice at six weeks. The results revealed that testes isolated from ASB17^−/−^ male mice had significantly fewer apoptotic cells than those from WT mice in both the control and experiment groups (Figure 4A,B). Furthermore, the intratesticular injection of etoposide significantly promoted germ cell apoptosis in both WT and ASB17^−/−^ testes (Figure 4A,B). The data showed that the apoptosis of ASB17^−/−^ testes decreased by 2.8-fold (*p* = 0.0001) in the control group and 2.4-fold (*p* = 0.0002) in the experiment group, respectively (Figure 4C). Moreover, much less cleaved-Caspase-3 protein expression could be detected in ASB17^−/−^ testes from both the control and experiment groups (Figure 4D). Overall, etoposide can promote germ cell apoptosis in both WT and ASB17^−/−^ mice, whereas ASB17^−/−^ mice were much more resistant in response to etoposide (ETO) treatment. Thus, it indicated that ASB17 deficiency prevented apoptosis of spermatogonia induced by etoposide in male mice.

### 3.5. ASB17 Promotes Apoptosis In Vitro

To further invest ASB17′s role in apoptosis, Hela cells were transfected with Flag-tagged ASB17. We chose cleaved (activated) PARP, cleaved Caspase-9, and cleaved Caspase-3 as the indicators of apoptosis [23]. The results showed that the cleaved PARP, cleaved Caspase-9, and cleaved Caspase-3 (Figure 5A) were elevated in the presence of ASB17. Since ASB17 was of low expression in Hela cells, we next engineered Hela cells stably expressed ASB17 and detected the mRNA level of ASB17 (Figure 5B). Staurosporine (STS) and etoposide (ETO) were used to induce apoptosis in Hela cells [24,25]. Firstly, a cell counting kit-8 (CCK8) assay was used to check the effect of ASB17 on the cell viability of Hela cells with or without STS and ETO. The results showed the number of cell survival in the control group was significantly higher than that in the experiment group (Figure 5C). The Western blot results showed the cleaved PARP, cleaved Caspase-9, and cleaved Caspase-3 were significantly elevated in ASB17 stably expressed Hela cells compared to the control group (Figure 5D,E). Furthermore, STS was added in a time-dependent manner to induce apoptosis. Following exposure to STS, the ectopic expression of ASB17 increased the cleaved PARP, cleaved Caspase-9, and cleaved Caspase-3 (Figure 5F). Wandering whether ASB17 promoted apoptosis in a caspase-dependent way, Z-VAD-FMK, an inhibitor of caspase, was used [26]. Following exposure to ETO and Z-VAD-FMK, ASB17 could no longer promote apoptosis (Figure 5G). Collectively, these results indicated that ASB17 promoted apoptosis in Hela cells in a caspase-dependent manner.

### 3.6. ASB17 Interacts with BCL2, BCLX, BCLW, and MCL1

To further determine the inner mechanism of ASB17, coimmunoprecipitation (Co-IP) assay was used to find the apoptosis-associated proteins that might interact with ASB17. The results revealed that ASB17 interacted with ATG7, BCLX, BCLW, MCL1, and BCL2 (Figure 6A). Among them, BCLX, BCLW, MCL1, and BCL2 attracted our interest. They are all BCL2 family members and important pro-survival proteins [27]. To verify the interaction between ASB17 and the four proteins, HEK293T cells were transfected with plasmids expressing BCL2, BCLX, BCLW, or MCL1 together with HA-tagged ASB17. The Co-IP results confirmed ASB17 interacted with BCL2, BCLX, BCLW, and MCL1 (Figure 6B–I). Furthermore, the immunofluorescence microscopy results showed that ASB17, BCL2, BCLX, BCLW, or MCL1 alone diffusely distributed in the cytoplasm, whereas ASB17 colocalized with BCL2, BCLX, BCLW, and MCL1 (Figure 7A–D). These results demonstrated that ASB17 interacted with BCL2, BCLX, BCLW, and MCL1.

### 3.7. ASB17 Promotes the Ubiquitylation and Degradation of BCLW and MCL1

The influence of ASB17 on the protein expression of BCL2, BCLX, BCLW, and MCL1 was checked. Notably, ASB17 was found to reduce the BCLW and MCL1 protein levels (Figure 8A). A Hela cell line stably expressing ASB17 was successfully generated. ASB17*,* BCLW*,* and MCL1 mRNA levels were detected (Figure 8B,D,F). Next, Flag-tagged BCLW or MCL1 were transfected into the control group and ASB17 stably expressed group of Hela cells. The results indicated ASB17 decreased the protein level of both BCLW and MCL1 but did not affect their mRNA level. (Figure 8C,E). Furthermore, ASB17 overexpression greatly decreased BCLW and MCL1 protein expression in a dose-dependent manner (Figure 8G,H). Cycloheximide (CHX) chase assay was used to compare the stability of HA-BCLW (Figure 8I) and HA-MCL1 (Figure 8J) with or without Flag-tagged ASB17. As expected, both HA-BLW (Figure 8I) and HA-MCL1 (Figure 8J) were destabilized by the ectopic expression of ASB17. The ASB17-mediated repression of BCLW (Figure 8K) and MCL1 (Figure 8L) was reversed by the proteasome inhibitor MG132. Moreover, mutant ASB17 lacking SOCS box (251–296aa) failed to decrease the protein level of BCLW (Figure 8M) and MCL1 (Figure 8N). ASB17 is a member of ASB family proteins, which show E3 ligase enzymatic activity [15]. We next examined the degradation of BCLW and MCL1 mediated by ASB17. Myc-tagged ASB17 and HA-ubiquitin were transfected into Hela cells together with Flag-tagged BCLW or MCL1. The results revealed that ASB17 promoted the ubiquitylation of BCLW and MCL1 (Figure 8O,P). Furthermore, the deletion of the SOCS box abolished the capability of ASB17 to promote the ubiquitylation of BCLW (Figure 8Q) and MCL1 (Figure 8R). Altogether, these data indicated that ASB17 promoted the ubiquitylation and degradation of BCLW and MCL1.

## 4. Discussion

E3 ubiquitin ligases have been found to play an important role in regulating testis development and spermatogenesis [28,29]. As a member of ASB family proteins, which shows E3 ubiquitin ligase enzymatic activity, ASB17, which is highly expressed in the testis, remains poorly understood. Here, we characterized the expression pattern and function of ASB17 to our best for the first time, helping to extend the knowledge in this field.

Unlike the previous study in which ASB17 was exclusively expressed in the testis, we found ASB17 was also expressed in the epididymis, spleen, and lung. The temporospatial expression pattern of ASB17 in mice was in line with what was reported before [17]. Compared with wild-type mice, ASB17^−/−^ mice had fewer apoptotic cells in the testes by using TUNNEL assays. We expected something different in testis development and spermatogenesis between WT and ASB17^−/−^ mice. Notably, no significant difference was detected in the testes’ weight and histology. ASB17^−/−^ mice at an early age remained normal in testis development and spermatogenesis. We thought that ASB17 might be not so important in the testis under normal conditions. However, under certain stimuli-like testicular toxins, heat stress, or chemotherapeutic agents that triggered germ cell apoptosis, ASB17 might help to activate apoptosis in the testis [30]. We then found that ASB17^−/−^ mice were much less sensitive to the stimuli of etoposide. It came to the conclusion that ASB17 did help activate apoptosis in the testis under etoposide stimuli.

We wondered how ASB17 promoted apoptosis and the mechanism inside it and then screened some apoptosis-associated proteins that might interact with ASB17. Among them, four BCL2 family proteins attracted our interest. We confirmed their interaction by Co-IP and immunofluorescence microscopy. Next, we checked if ASB17 affected the protein level of the four proteins and found that ASB17 degraded BCLW and MCL1 in a ubiquitylation-dependent manner. These results confirmed that ASB17 could promote apoptosis. Since ASB17 can degrade BCLW and MCL1, the function of the two proteins can broaden our knowledge about ASB17. Even though the mice that lacked BCLW were normal, male BCLW-deficient mice were infertile and had much more apoptotic cells in the testes than those from the control group [31]. Since BCLW was located in the round and elongating spermatids where ASB17 was located. This might explain why male ASB17^−/−^ mice had fewer apoptotic cells than those from WT mice. BCLW was also found to be elevated in Alzheimer’s disease (AD) and had a protective role in neurons [32,33]. It meant ASB17 could have a role in AD too. MCL1 is highly expressed in a variety of human cancer and is often related to chemotherapeutic resistance and relapse [34]. E3 ubiquitin ligases (Mule, SCF^β-TrCP^, SCF^FBW7^, TRIM17, APC/C^Cdc20^, and FBXO4) and deubiquitinases (USP9X, Ku70, USP13, JOSD1, and DUB3) work together to balance MCL1 stability, which has an important role in the chemoresistance of cancer cells [35,36]. Our finding that that ASB17 promoted the ubiquitylation of MCL1 and degraded it might provide a potential therapeutic option. Because no antibody of good quality for ASB17 can be used, a lot of endogenous experiments between ASB17 and BCLW or MCL1 were unable to be undertaken. Overall, our findings suggested that E3 ubiquitin ligase ASB17 was a positive regulator of cell apoptosis by promoting ubiquitylation and degradation of BCLW and MCL1. Furthermore, we provided a potential target for BCLW- and MCL1-related diseases.

## 5. Conclusions

In this study, we elaborated the function of ASB17 in apoptosis pathway. ASB17 was mainly expressed in testis and promoted apoptosis both in vivo and in vitro. Furthermore, ASB17 deficiency resulted in the reduction of apoptosis in spermatogenic cells, but it did not affect the development of spermatozoa or the normal fertility. And the ASB17^−/−^ mice were more resistant to the stimuli of etoposide. Next, overexpression of ASB17 could promote apoptosis in HELA cells. Moreover, BCLW and MCL1 were found to interact with ASB17. And, ASB17 can promote ubiquitylation and degradation of them. Overall, ASB17 was a novel positive regulator of cell apoptosis.

## Figures and Tables

**Figure 1 biology-10-00234-f001:**
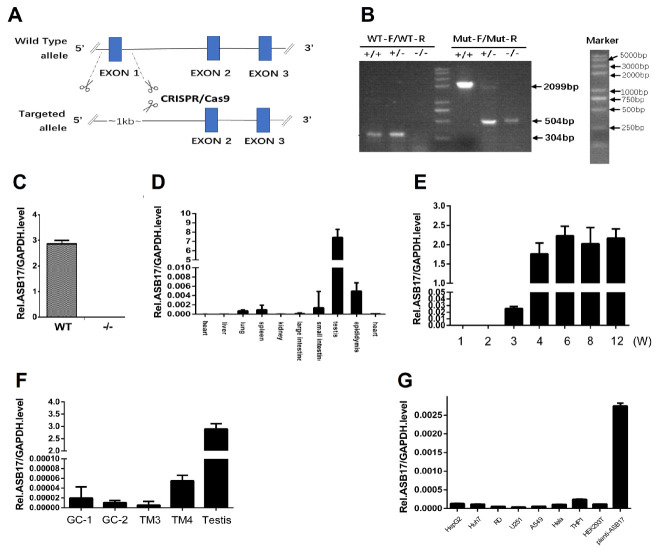
Generation of ASB17^−/−^ mice and the expression pattern of ASB17. (**A**) Schematic diagram of the CRISPR/Cas9 targeting technology: exon1 of ASB17 was deleted. (**B**) Genotype of ASB17^+/+^, ASB17^+/−^, and ASB17^−/−^ mice were examined by two pairs of primers using genome DNA isolated from tails snips. (**C**) Quantitative real-time PCR (Q-PCR) analysis of ASB17 in genotype ASB17^+/+^ and ASB17^−/−^ testes. (**D**) Q-PCR analysis of ASB17 in different organs from ASB17^+/+^ mouse. (**E**) Q-PCR analysis of ASB17 in the testes of different ages from ASB17^+/+^ mice. (**F**) Q-PCR analysis of ASB17 in 4 testis-derived cell lines (GC-1, GC-2, TM3, and TM4). (**G**) Q-PCR analysis of ASB17 in eight usual cell lines (HepG2, Huh7, RD, U251, A549, Hela, THP1, and 293T) together with lentivirus-carrying ASB17-vector-infected Hela cells.

**Figure 2 biology-10-00234-f002:**
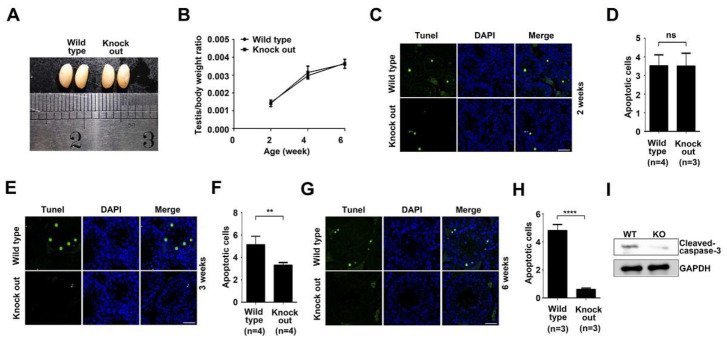
ASB17 deficiency in the testes decreases apoptosis but does not affect testes development. (**A**) Testes isolated from ASB17^+/+^ and ASB17^−/−^ mice at the age of 6 weeks. (**B**) Testis/body weight ratio of ASB17^+/+^ and ASB17^−/−^ mice at the ages of 2, 4, and 6 weeks (*n* = 3). (**C**,**E,G**) TUNEL assay was used to check the apoptotic cells in the testes of ASB17^+/+^ and ASB17^−/−^ mice at the ages of 2 (**C**), 3 (**E**), and 6 weeks (**G**). TUNEL (green) and DAPI (blue) were observed. Bar = 50 μm. (**D,F,H**) Quantification and comparison of apoptotic cells in the ASB17^+/+^ and ASB17^−/−^ testes at 2 (**D**), 3 (**F**), and 6 weeks (**H**). (**I**) Western blot analysis of cleaved-Caspase-3 protein expression in the testes from wild-type (WT) and ASB17^−/−^ mice. GAPDH was used as a loading control. For each mouse, 100 random seminiferous tubules sections were analyzed to count the apoptotic cells, and per 10 random tubules, random tubules cross-sections were presented as a number. Data shown are the mean ± SEM. **, *p* < 0.01; ****, *p* < 0.0001, and no significance (ns).

**Figure 3 biology-10-00234-f003:**
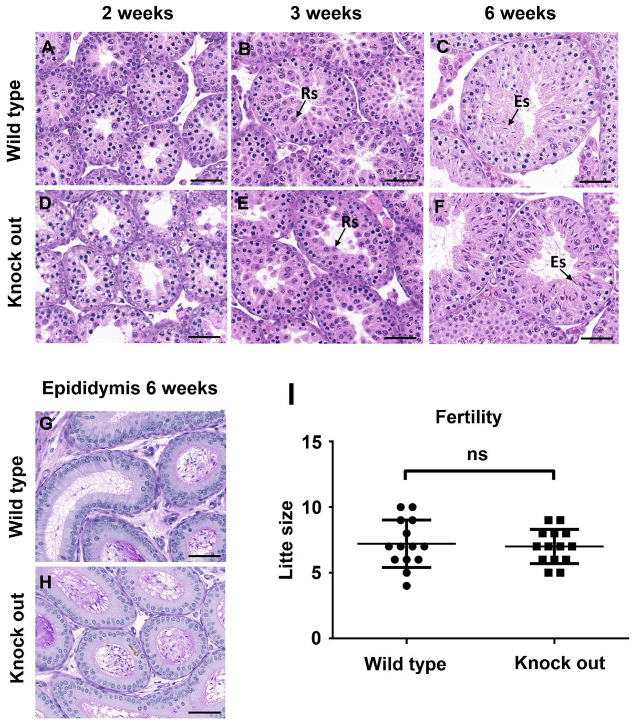
ASB17^−/−^ mice show normal spermatogenesis. (**A**–**F**) Histology of testes isolated from ASB17^+/+^ and ASB17^−/−^ mice at the ages of 2, 3, and 6 weeks were revealed by hematoxylin and eosin (H&E). Arrows point to round spermatids (Rs) and elongated spermatids (Es). Bar = 50 μm. (**G**,**H**) Representative lumen of epididymis sections from wild-type (**G**) and ASB17^−/−^ (**H**) mice stained with periodic acid–Schiff (PAS). Bar = 50 μm. (**I**) Fertility evaluation of the wild-type and ASB17^−/−^ males, with bars indicating SEM.

**Figure 4 biology-10-00234-f004:**
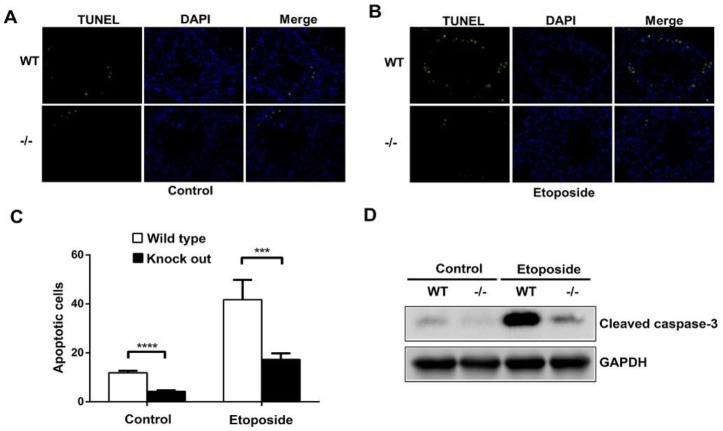
ASB17 deficiency prevents the apoptosis of spermatogonia induced by etoposide in male mice. (**A**,**B**) TUNEL assay was used to check the apoptotic cells in the testes of ASB17^+/+^ and ASB17^−/−^ mice in control (*n* = 4) (**A**) and etoposide-induced group (*n* = 5) (**B**). TUNEL (green) and DAPI (blue) were observed. Bar = 50 μm. (**C**) Quantification and comparison of apoptotic cells of the ASB17^+/+^ and ASB17^−/−^ testes in the control and etoposide-induced groups. (**D**) Western blot analysis of cleaved-Caspase-3 protein expression in the testes from WT and ASB17^−/−^ mice of the control and experiment groups. GAPDH was used as a loading control. For each mouse, 100 random seminiferous tubules sections were analyzed to count the apoptotic cells, and per 10 random tubules, random tubules cross-sections were presented as a number. Data shown are the mean ± SEM. ***, *p* < 0.01; ****, *p* < 0.0001.

**Figure 5 biology-10-00234-f005:**
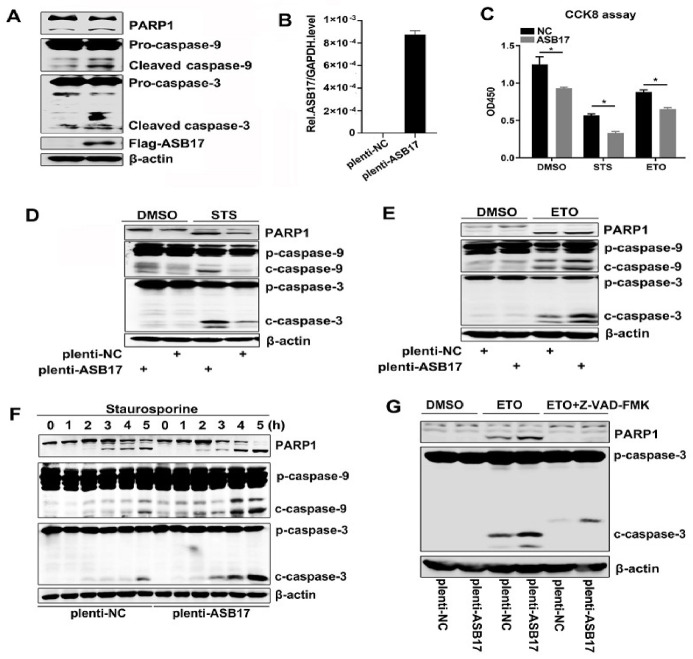
ASB17 promotes apoptosis in vitro. (**A**) Immunoblot analysis of extracts from Hela cells transfected with control plasmid or Flag-ASB17. The cell lysates were immunoblotted with the indicated antibodies. (**B**) Q-PCR analysis of mRNA of ASB17 of control and ASB17 stably expressed Hela cells. (**C**) CCK8 assay of control and ASB17 stably expressed Hela cells treated by DMSO, staurosporine (STS) (1 μM) for 4 h, and etoposide (ETO) (1 μg/mL) for 24 h. (**D**,**E**) The control and ASB17 stably expressed Hela cells were treated with DMSO and STS (**D**) for 4 h or ETO (**E**) for 24 h. Cell lysates were immunoblotted with the indicated antibodies. (**F**) Immunoblot analysis of extracts from control and ASB17 stably expressed Hela cells treated with STS in the indicated time. (**G**) Immunoblot analysis of extracts from control and ASB17 stably expressed Hela cells treated with DMSO, ETO, or ETO and Z-VAD-FMK for 24 h. *, *p* < 0.05.

**Figure 6 biology-10-00234-f006:**
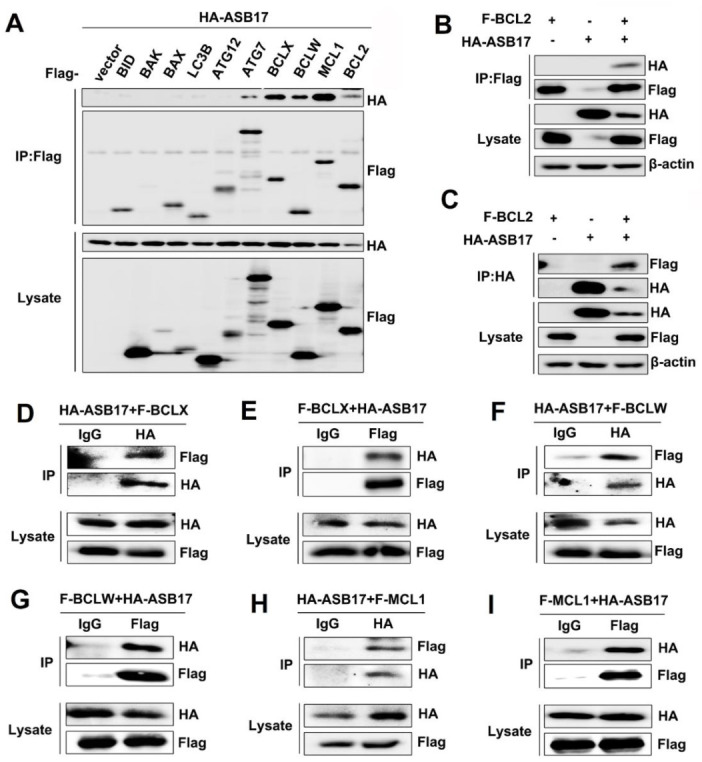
ASB17 interacts with BCL2, BCLX, BCLW, and MCL-1. (**A**) HEK293T cells were transiently co-transfected with the control plasmids, Flag-BID, Flag-BAK, Flag-BAX, Flag-LC3B, Flag-ATG12, Flag-ATG7, Flag-BCLX, Flag-BCLW, Flag-MCL1, or Flag-BCL2 together with HA-ASB17. The cell lysates were immunoprecipitated with an anti-Flag antibody followed by anti-HA immunoblotting. (**B**,**C**) HEK293T cells were transiently co-transfected with HA-ASB17 and control plasmids, Flag-BCL2 and control plasmids, or HA-ASB17 and Flag-BCL2 plasmids, respectively, and analyzed by immunoprecipitation (IP) with an anti-Flag (**B**) or anti-HA antibody (**C**) and immunoblotting (IB) with indicated antibodies. (**D**,**E**) HEK293T cells were transiently co-transfected with HA-ASB17 and Flag-BCLX. The cell lysates were immunoprecipitated with an anti-IgG and anti-HA (**D**) or anti-Flag (**E**) antibody followed by immunoblotting with indicated antibodies. (**F**,**G**) HEK293T cells were transiently co-transfected with HA-ASB17 and Flag-BCLW. The cell lysates were immunoprecipitated with an anti-IgG and anti-HA (**F**) or anti-Flag (**G**) antibody followed by immunoblotting with indicated antibodies. (**H**,**I**) HEK293T cells were transiently co-transfected with HA-ASB17 and Flag-MCL1. The cell lysates were immunoprecipitated with an anti-IgG and anti-HA (**H**) or anti-Flag (**I**) antibody followed by immunoblotting with indicated antibodies.

**Figure 7 biology-10-00234-f007:**
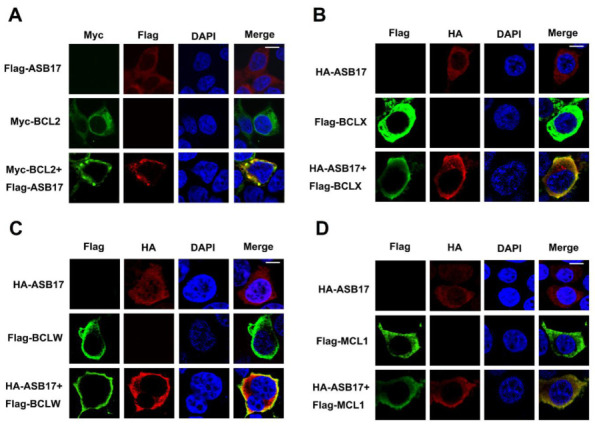
ASB17 interacts with BCL2, BCLX, BCLW, and MCL-1. (**A**) HEK293T cells were co-transfected with Flag-ASB17 and control plasmids, Myc-BCL2 and control plasmids, or Flag-ASB17 and Myc-BCL2 plasmids, respectively. Subcellular localizations of Myc-BCL2 (green), Flag-ASB17 (Red), and nucleus marker DAPI (blue) were analyzed under confocal microscopy. Bar = 10 μm. (**B**) HEK293T cells were co-transfected with HA-ASB17 and control plasmids, Flag-BCLX and control plasmids, or HA-ASB17 and Flag-BCLX plasmids, respectively. Subcellular localizations of Flag-BCLX (green), HA-ASB17 (Red), and nucleus marker DAPI (blue) were analyzed under confocal microscopy. Bar = 10 μm. (**C**) HEK293T cells were co-transfected with HA-ASB17 and control plasmids, Flag-BCLW and control plasmids, or HA-ASB17 and Flag-BCLW plasmids, respectively. Subcellular localizations of Flag-BCLW (green), HA-ASB17 (Red), and nucleus marker DAPI (blue) were analyzed under confocal microscopy. Bar = 10 μm. (**D**) HEK293T cells were co-transfected with HA-ASB17 and control plasmids, Flag-MCL1 and control plasmids, or HA-ASB17 and Flag-MCL1 plasmids, respectively. Subcellular localizations of Flag-MCL1 (green), HA-ASB17 (Red), and nucleus marker DAPI (blue) were analyzed under confocal microscopy. Bar = 10 μm.

**Figure 8 biology-10-00234-f008:**
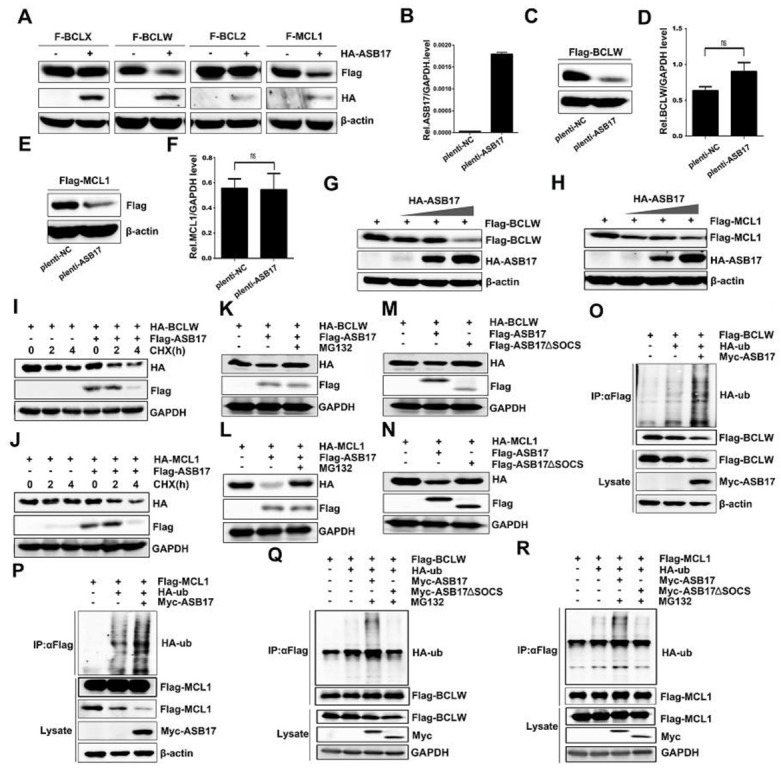
ASB17 promotes the ubiquitylation and degradation of BCLW and MCL1. (**A**) Immunoblot analysis of extracts from Hela cells transfected with Flag-BCLX, Flag-BCLW, Flag-BCL2, or Flag-MCL1 together with the control plasmids or HA-ASB17. The lysates were immunoblotted with indicated antibodies. (**B**,**D**,**F**) Q-PCR analysis of mRNA of ASB17 (**B**), BCLW (**D**), and MCL1 (**F**) in ASB17 stably expressed and control Hela cells, with bars indicating SEM. (**C**,**E**) Immunoblot analysis of extracts from the control and ASB17 stably expressed Hela cells transfected with Flag-BCLW (**C**) or Flag-MCL1 (**E**). (**G**,**H**) Flag-BCLW (**G**) or Flag-MCL1 (**H**) and an increasing dose of HA-ASB17 were transfected into Hela cells. The cell lysates were immunoprecipitated with indicated antibodies. (**I**,**J**) Hela cells transfected with HA-BCLW (**I**) or HA-MCL1 (**J**) in the presence or absence of F-ASB17 were exposed to 50 μg/mL of CHX for 2 and 4 h. The lysates were immunoprecipitated with antibodies against HA, Flag, and GAPDH. (**K**,**L**) Hela cells were co-transfected with HA-BCLW (**K**) or HA-MCL1 (**L**) together with the control plasmids or F-ASB17. Cells were then treated with DMSO or MG132 (10 μM). The lysates were immunoblotted with indicated antibodies. (**M**,**N**) Immunoblot analysis of extracts from Hela cells transfected with HA-BCLW (**M**) or HA-MCL1 (**N**) together with the control plasmids or F-ASB17 ΔSOCS. The lysates were immunoblotted with indicated antibodies. (**O**,**P**) Flag-BCLW (**O**) or Flag-MCL1 (**P**), HA-Ub, and Myc-ASB17 plasmids were transfected into Hela cells. The cell lysates were immunoprecipitated with anti-Flag antibody and then immunoblotted with indicated antibodies. (**Q**,**R**) Flag-BCLW (**Q**) or Flag-MCL1 (**R**), HA-Ub, and Myc-ASB17 ΔSOCS plasmids were transfected into Hela cells. The cell lysates were immunoprecipitated with anti-Flag antibody and then immunoblotted with indicated antibodies.

## Data Availability

All data needed to evaluate the conclusions in the paper are present in the paper.

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
