# Peer review of "E3 Ubiquitin Ligase ASB17 Promotes Apoptosis by Ubiquitylating and Degrading BCLW and MCL1"

_biology, 2021, doi:10.3390/biology10030234_

Round 1

Reviewer 1 Report

Comments to authors

In this manuscript, Yang and colleagues identify the E3 ubiquitin ligase ASB17 as a critical regulator of apoptosis, especially in testis. By generating ASB17 KO mice, the author show that ABS17 is required for testicular apoptosis. Overexpression studies using HeLa cells reveal that ABS17 slightly increases STS- or ETO-induced apoptosis with a decrease in BCLW and MCL1 protein levels. Finally, the author show that ASB17 increases BCLW and MCL1 ubiquitination and subsquent degradation. While the identification of a new E3 ligase that regulate apoptosis is higly interesting in the field and the key observations are clear and convincing, there are a few key experiments needed to further support their clames.

Major points

  1. In Fig. 2 and 4, TUNEL staining is no longer a specific apoptosis marker. Therefore, other apoptotic markers such as cleaved-caspase-3 (using IHC or western blot) or Annexin V staining (using Flow cytometry) should be supplemented.
  2. In Fig. 5, the cleaved-PARP and cleaved-caspases levels only slightly increased during ASB17 ectopic expression. Therefore, another cell death assay is needed. Alternatively, the authors can use the pan-caspase inhibitor zVAD-fmk to support that the observed cell death is apoptosis.
  3. In Fig. 8A-F, while ASB17 induces downregulation of BCLX and MCL1, there is no evidence for protein degradation. First, mRNA levels of BCLX and MCL1 should be provided. Second, protein stability data measured using CHX are also required. Alternatively, the authors can employ proteasomal inhibitor such as MG132 or lysosomal inhibitor to determine which degradation pathway is involved in the downregulation of BCLW and MCL1.
  4. In Fig. 8E-H, experiment using the E3 defective mutant is needed to show whether E3 ligase activity is indeed required for the ubiquitination and degradation process.
  5. In Fig. 8G, H, immunoprecipitated BCLW and MCL1 are required as an IP control. If these proteins are degraded by the proteasome, MG132 may increase the ubiquitination efficiency.
  6. Most of the data provided in the manuscript are performed using proteins overexpressed in cells, which means ASB17 has the ability to increase ubiquitination and degradation of BCLW and MCL1, but it is unknown wether this effect is direct or not. Therefore, it is not yet known to what extent ASB17 contributes to the ubiquitination of these proteins in a physiological context before showing the ubiquitination of endogenous proteins under ASB17 depletion or depletion.
  7. Furthermore, without in vitro ubiquitination data, it is difficult to conclude that ASB17 is an E3 ligase for those protein. If there are technical difficulties, sush as low endogenous expression levels and low antibody specificity, the authors should dicuss the limitations of this study using overexpressed proteins.

Author Response

Reviewer 1

  1. In Fig. 2 and 4, TUNEL staining is no longer a specific apoptosis marker. Therefore, other apoptotic markers such as cleaved-caspase-3 (using IHC or western blot) or Annexin V staining (using Flow cytometry) should be supplemented.

Authors’ Response: Thank you for the suggestion.

As you suggested, we have performed the related experiments and the new results have been shown in Revised Figure 2I and Revised Figure 4D in the revised manuscript.

  1. In Fig. 5, the cleaved-PARP and cleaved-caspases levels only slightly increased during ASB17 ectopic expression. Therefore, another cell death assay is needed. Alternatively, the authors can use the pan-caspase inhibitor zVAD-fmk to support that the observed cell death is apoptosis.

Authors’ Response: Thank you for the comment.

The CCK8 assay in Figure 5C can help to prove our conclusion that ASB17 promoted apoptosis in vitro. According to your suggestion, we have performed additional experiment to support that the observed cell death is apoptosis by using the pan-caspase inhibitor Z-VAD-FMK in revised Figure 5G in the revised manuscript.

  1. In Fig. 8A-F, while ASB17 induces downregulation of BCLX and MCL1, there is no evidence for protein degradation. First, mRNA levels of BCLX and MCL1 should be provided. Second, protein stability data measured using CHX are also required. Alternatively, the authors can employ proteasomal inhibitor such as MG132 or lysosomal inhibitor to determine which degradation pathway is involved in the downregulation of BCLW and MCL1.

Authors’ Response: Thank you for the comment.

As you suggested, we have carried out additional experiments to provide evidence for protein degradation. The new results have been shown in revised Figure 8D and F–L  in the revised manuscript.

  1. In Fig. 8E-H, experiment using the E3 defective mutant is needed to show whether E3 ligase activity is indeed required for the ubiquitination and degradation process.

Authors’ Response: Thank you again.

As you suggested, we have carried out additional experiments to provide evidence for protein degradation. The new results have been shown in revised Figure 8M, N, Q, and R in the revised manuscript.

  1. In Fig. 8G, H, immunoprecipitated BCLW and MCL1 are required as an IP control. If these proteins are degraded by the proteasome, MG132 may increase the ubiquitination efficiency.

Authors’ Response: Again, thank you.

According to your suggestion, we have performed additional experiments by adding immunoprecipitated BCLW and MCL1. The new results have been shown in revised Figure 8O and P. The data indicated that MG32 did increase the ubiquitination efficiency.

  1. Most of the data provided in the manuscript are performed using proteins overexpressed in cells, which means ASB17 has the ability to increase ubiquitination and degradation of BCLW and MCL1, but it is unknown if this effect is direct or not. Therefore, it is not yet known to what extent ASB17 contributes to the ubiquitination of these proteins in a physiological context before showing the ubiquitination of endogenous proteins under ASB17 depletion or depletion.

Authors’ Response: Thank you for the question.

It is a good suggestion. However, we have had a very difficult time to isolate primary sperm cells from mice.

  1. Furthermore, without in vitroubiquitination data, it is difficult to conclude that ASB17 is an E3 ligase for those protein. If there are technical difficulties, such as low endogenous expression levels and low antibody specificity, the authors should discuss the limitations of this study using overexpressed proteins.

Authors’ Response: Thank you again.

We demonstrated that overexpressed ASB17 could promote apoptosis. However, we did not know in which primary cell types ASB17 could promote ubiquitylation and degradation of BCL-w and MCL-1.  

Reviewer 2 Report

This manuscript written by Yang, Wan et al. focuses on the examination of ASB17, an ankyrin-repeat containing E3 ubiquitin ligase that is predominantly expressed in the testes. Specifically, the authors used CRISPR/Cas9 to create ASB17 knockout mice, histology and immunological staining, and CoIP assays to demonstrate that ASB17 plays a role in promoting apoptosis of spermatogonia.  The researchers also convincingly demonstrate that ASB17 interacts with members of the BCL2 family and that ASB17-dependent ubiquitylation of BCL-w and MCL-1 is required for apoptosis.  The quality of the images and proper use of controls throughout all of their experiments show that the authors took great care in their experimental design and analysis.

Major concern:
Numerous grammatical mistakes throughout manuscript that need to be addressed.  Many colloquial phrases and tense shifts in text also need to be fixed.

Minor concerns:
Since the attachment of ubiquitin to a protein is a posttranslational modification, the correct term that should be used throughout this manuscript is “ubiquitylation”, “ubiquitylate”, and/or “ubiquitylated”.  This is analogous to phosphorylation, methylation, acetylation, acylation, etc.

All acronyms should be defined as soon as they are introduced in the text.

Author Response

Reviewer 2

Major concern:

Numerous grammatical mistakes throughout manuscript that need to be addressed.  Many colloquial phrases and tense shifts in text also need to be fixed.

Authors’ Response: Thank you for the comment.

According to your suggestion, we have checked the grammatical mistakes in the revied manuscript carefully. The colloquial phrases and tense shifts have been corrected in revised manuscript.

Minor concerns:

Since the attachment of ubiquitin to a protein is a posttranslational modification, the correct term that should be used throughout this manuscript is “ubiquitylation”, “ubiquitylate”, and/or “ubiquitylated”.  This is analogous to phosphorylation, methylation, acetylation, acylation, etc.

All acronyms should be defined as soon as they are introduced in the text.

Authors’ Response: Thank you for the comment.

According to your suggestion, the correct term related to ubiquitin has been used. The acronyms have been defined as they were introduced in the text of the revised manuscript.

Reviewer 3 Report

The manuscript demonstrated potential apoptosis-promoting role of ASB17 in spermatogenesis by degrading BCL-w and MCL-1 in mice. The results are interesting; however, there seems to be several serious problems, that failed to give scientific accuracy to this work.

1) The current study is not the first work reporting the testicular expression of ASB17. It was already reported by Guo et al. in 2004 (Arch Androl 2004;50:155-161). The authors should not neglect the previous work and should cite the article as a reference.

2) The most unconvincing point in this study is that the authors did not use any proteasome inhibitors at all. I never believe that researchers in general cannot visualize IP-Western blot image of poly-ubiquitinated proteins without using proteasome inhibitors, including ALLN and MG-132. Without such agents, it may be impossible to obtain Figures 8 G and H. Likewise, how the authors could get confocal images of 2-protein expression without using proteasome inhibitors? According to the authors’ scenario, the E3 ubiquitin ligase ASB17 should promote degradation of BCL-w (Fig. 7C) and MCL-1 (Fig. 7D). This means that it is difficult to simultaneously observe the expression of the two proteins under ordinary condition. In addition, nobody can visualize direct interaction of two tagged proteins in IP-Western experiments without using the agents (Fig. 6).

3) Etoposide experiment (Fig. 4) failed to show significant decrease in etoposide-induced apoptosis in ASB17-deficient mice. Under untreated condition, the decrease in apoptosis was found in ASB17-deficient mice. Was the decrease further enhanced by the treatment with etoposide?         

4) The authors should observe whether loss of ASB17 involves in hyperplasia and/or carcinogenesis in testis and discuss the fate.

5) The E3 ligase SCFFBW7 is known to target MCL1 (Nature 2011;471: 104-109). The authors should discuss ASB17 as another potential E3 ligase toward MCL1.

Author Response

Reviewer 3

  1. The current study is not the first work reporting the testicular expression of ASB17. It was already reported by Guo et al. in 2004 (Arch Androl 2004;50:155-161). The authors should not neglect the previous work and should cite the article as a reference.

Authors’ Response: Thank you for the comment.

According to your suggestion, we have cited the article as a reference in the revised manuscript.

  1. The most unconvincing point in this study is that the authors did not use any proteasome inhibitors at all. I never believe that researchers in general cannot visualize IP-Western blot image of poly-ubiquitinated proteins without using proteasome inhibitors, including ALLN and MG-132. Without such agents, it may be impossible to obtain Figures 8 G and H. Likewise, how the authors could get confocal images of 2-protein expression without using proteasome inhibitors? According to the authors’ scenario, the E3 ubiquitin ligase ASB17 should promote degradation of BCL-w (Fig. 7C) and MCL-1 (Fig. 7D). This means that it is difficult to simultaneously observe the expression of the two proteins under ordinary condition. In addition, nobody can visualize direct interaction of two tagged proteins in IP-Western experiments without using the agents (Fig. 6).

Authors’ Response: Thank you again.

As you suggested, we have performed additional experiments by adding MG132. The new results have been shown in revised Figure 8K, L, Q, and R, and which indicated that it MG132 could increase the ubiquitination efficiency. Some articles did not use  proteasome inhibitors when doing Co-IP, or ubiquitination experiments, such as “E3 ligase WWP2 negatively regulates TLR3-mediated innate immune response by targeting TRIF for ubiquitination and degradation”[1], “The E3 ubiquitin ligase TRIM31 attenuates NLRP3 inflammasome activation by promoting proteasomal degradation of NLRP3”[2], and “E3 ubiquitin ligase tripartite motif 38 negatively regulates TLR-mediated immune responses by proteasomal degradation of TNF receptor-associated factor 6 in macrophages”[3], etc. In my opinion, BCLW and MCL1 together with ASB17 were overexpressed in our experiments. Even ASB17 could degrade BCLW or MCL1, there were some proteins left that can be detected. And we sometimes transfected less ASB17 plasmid to easier detect BCLW or MCL1.

  1. Etoposide experiment (Fig. 4) failed to show significant decrease in etoposide-induced apoptosis in ASB17-deficient mice. Under untreated condition, the decrease in apoptosis was found in ASB17-deficient mice. Was the decrease further enhanced by the treatment with etoposide?

Authors’ Response: Thank you for the comment.

Our results indicated that the decrease was not enhanced by the treatment with etoposide.   

  1. The authors should observe whether loss of ASB17 involves in hyperplasia and/or carcinogenesis in testis and discuss the fate.

Authors’ Response: Thank you again for the comment.

Based on our experiments, we found that the testes from ASB17-/- mice were not involved in hyperplasia and/or carcinogenesis. ASB17 might be not so important in testis under normal conditions. But under certain stimuli like testicular toxins, heat stress, or chemotherapeutic agents that triggered germ cells apoptosis, ASB17 might help to activate apoptosis in testis.

  1. The E3 ligase SCFFBW7 is known to target MCL1 (Nature 2011;471: 104-109). The authors should discuss ASB17 as another potential E3 ligase toward MCL1.

Authors’ Response: Thank you again.

We have already discussed ASB17 as another potential E3 ligase toward MCL1, which began from “MCL1 is highly expressed in a variety of human cancer…” and we have cited the article as a reference in the revised manuscript.

Reference

References

  1. Yang, Y.; Liao, B.; Wang, S.; Yan, B.; Jin, Y.; Shu, H.B.; Wang, Y.Y. E3 ligase WWP2 negatively regulates TLR3-mediated innate immune response by targeting TRIF for ubiquitination and degradation Proc Natl Acad Sci U S A, 2013, 110, 5115-5120.
  2. Song, H.;Liu, B.;Huai, W.;Yu, Z.;Wang, W.;Zhao, J.;Han, L.;Jiang, G.;Zhang, L.;Gao, C.; et al. The E3 ubiquitin ligase TRIM31 attenuates NLRP3 inflammasome activation by promoting proteasomal degradation of NLRP3 Nat Commun, 2016, 7, 13727.
  3. Zhao, W.; Wang, L.; Zhang, M.; Yuan, C. Gao, C. E3 ubiquitin ligase tripartite motif 38 negatively regulates TLR-mediated immune responses by proteasomal degradation of TNF receptor-associated factor 6 in macrophages J Immunol, 2012, 188, 2567-2574.

Round 2

Reviewer 1 Report

All responses are fine, but I don't see revised figure in the manuscript while the text and figure legends are revised.